# Omics Approaches to Assess Flavor Development in Cheese

**DOI:** 10.3390/foods11020188

**Published:** 2022-01-11

**Authors:** Rania Anastasiou, Maria Kazou, Marina Georgalaki, Anastasios Aktypis, Georgia Zoumpopoulou, Effie Tsakalidou

**Affiliations:** Laboratory of Dairy Research, Department of Food Science and Human Nutrition, Agricultural University of Athens, Iera Odos 75, 118 55 Athens, Greece; kmaria@aua.gr (M.K.); mgeor@aua.gr (M.G.); aktyp@aua.gr (A.A.); gz@aua.gr (G.Z.); et@aua.gr (E.T.)

**Keywords:** cheese, flavor, omics, lactic acid bacteria, yeasts, cheese microbiome, metagenomics, metatranscriptomics, metabolomics

## Abstract

Cheese is characterized by a rich and complex microbiota that plays a vital role during both production and ripening, contributing significantly to the safety, quality, and sensory characteristics of the final product. In this context, it is vital to explore the microbiota composition and understand its dynamics and evolution during cheese manufacturing and ripening. Application of high-throughput DNA sequencing technologies have facilitated the more accurate identification of the cheese microbiome, detailed study of its potential functionality, and its contribution to the development of specific organoleptic properties. These technologies include amplicon sequencing, whole-metagenome shotgun sequencing, metatranscriptomics, and, most recently, metabolomics. In recent years, however, the application of multiple meta-omics approaches along with data integration analysis, which was enabled by advanced computational and bioinformatics tools, paved the way to better comprehension of the cheese ripening process, revealing significant associations between the cheese microbiota and metabolites, as well as their impact on cheese flavor and quality.

## 1. Introduction

The history of cheesemaking is lost in the mists of time [1]. Over the years, cheese making has greatly evolved with a remarkable diversity of cheeses produced worldwide, with culture and resources shaping the production in each region. Nowadays, the consumer demand for high-quality products with excellent sensorial characteristics and, at the same time, at a reasonable cost is growing. According to recent estimations, global cheese consumption is expected to increase by 13% from 2016 to 2025 [2].

Cheese flavor together with texture and overall appearance are crucial parameters that determine consumer preference and enjoyment [3,4]. In almost all cheese varieties, flavor develops mainly during the ripening process, a complex and time-dependent process that involves microbiological and biochemical changes mediated by the metabolic flux of starter, adjunct, and non-starter cultures, the action of enzymes from rennet and milk [5], together with non-enzymatic conversions [6]. Cheese manufacture and ripening are characterized by a succession of microbial communities that contribute via their metabolism to the production of numerous compounds, which provide aroma (odor) and flavor (taste) to cheese products [7]. Starter lactic acid bacteria (SLAB), carefully selected and deliberately added to milk, rapidly metabolize milk lactose, and produce lactic acid, thus improving the microbial safety and quality of the final product by readily acidifying the curd [8]. However, throughout ripening, SLAB decline, releasing intracellular enzymes due to autolysis, thus producing molecules beneficial for the growth of non-starter LAB (NSLAB) that take over, since they can grow on a food ecosystem that contains lactate, citrate, glycerol, ribose, free fatty, and amino acids [9]. NSLAB significantly contribute to cheese flavor, texture, nutritional value, and microbial safety in most of the ripened cheese varieties [10,11]; however, some cheese quality defects [9,12] and off-flavors, especially in the later phases of ripening, have been attributed to NSLAB [11,13,14]. Moreover, the de-acidifying, proteolytic, and/or lipolytic activities of yeasts and filamentous fungi significantly impact the development of cheese flavor, texture, and typical appearance, especially of smear- and mold-ripened cheeses, respectively [15,16,17].

Study of the complex microbial consortia dynamics and evolution during cheese making and ripening is critical in understanding their contribution to the sensorial characteristics of the final product. Recent advances in high-throughput sequencing (HTS) methodologies (e.g., amplicon metagenomics, shotgun metagenomics, and metatranscriptomics) and high-resolution metabolomics coupled with the constantly evolving bioinformatics and statistical tools for data analysis, as well as the improvement of available databases, have provided deeper insights into cheese microbiota diversity and interactions and thus revealed the impact of microbial metabolism on flavor compounds formation [18,19,20,21,22]. Furthermore, the integration of multiple meta-omics tools, together with powerful statistical analysis methods, has allowed detailed functional characterization of the cheese microbiota and determination of its contribution to the development of characteristic flavor metabolites [23,24,25,26,27,28,29]. This in-depth study of cheese microbiota could result in the application of selected cultures able to enhance or diversify cheese flavor, and thus, leading to cheese making practices and technologies for accelerated or improved ripening with economic benefit for both cheese producers and consumers.

In this review article, we first discuss the sources of cheese microbiota and the taxa as detected by both culture-dependent and -independent approaches, as well as the key microbial metabolic pathways that determine flavor formation. We then focus on studies where individual -omics technologies have been applied, and finally, on integrated multi-omics studies that unravel the composition and functionality of the complex cheese microbiota on flavor formation.

## 2. The Cheese Microbiome

### 2.1. Origin of the Cheese Microbiota

Cheese is a high-nutrient fermented dairy food that consists of proteins, sugars, fat, minerals, and vitamins, while its microbial consortium contributes to an added nutrient and even probiotic value by its metabolic activity. Knowledge of the cheese microbiome is important, as the microorganisms and their primarily biochemical processes will drastically affect the final product [30]. Cheese microbiota originates from three sources: the indigenous milk microbiota, the inoculated starter cultures, and the cheese production environment [26,31]. The mammalian species seems to influence milk microbiota diversity, with cow’s milk appearing to be more diverse than that of, e.g., goats and sheep. Moreover, the fat content also affects the microbial load [32,33,34]. The milk of a healthy lactating animal is not, although often considered, sterile, as bacteria isolates present in the mammary gland have been found to be other than those found on the skin of the same host [35]. In addition, the ability of some microbes to travel from the intestinal lumen to the mammary gland through an entero-mammary pathway has also been described [35,36]. More than 100 genera and 400 microbial species have been detected in raw cow, sheep, and goat milk, and in particular, 90 species of Gram-negative bacteria, 90 species of Gram-positive and catalase-positive bacteria, 60 species of lactic acid bacteria (LAB), and 70 and 40 species of yeasts and molds, respectively [31,37]. For all microbial groups, inter-farm variability is wide, while intra-farm variability is generally much lower except from season to season [38].

### 2.2. Factors Affecting the Cheese Microbiota

The load and composition of the raw milk microbiota depend directly on the hygiene conditions of the milking environment and sources that are in contact with the milk, such as the animal’s teat, milker’s hands, milking machine, milk line, and tank [39,40]. Crucial factors affecting the microbial dynamics are also the type of forage fed to animals and the grazing system employed, the feeding practices, and the overall management system of the farm including hygiene conditions and drinking and washing water [31,41,42]. Moreover, udder and teat cleaning as well as disinfection of the milking equipment are of significant importance as well [43,44].

Nowadays, several approaches are used to decrease the microbial load, thus ensuring milk safety, as the industrialized production process must meet strict sanitation and quality standards to ensure cheese safety [45]. Heat treatment and microfiltration are the most commonly adopted practices in both small- and large-scale dairy units to reduce the microbial load of milk. Moreover, from the initial step of milking to the shipment of cheeses to consumers, the product is subjected to rigorous monitoring, quality assurance standards, and specific microbiological tests. However, milk heat treatment not only reduces the spoilage and pathogenic population, but also the indigenous pro-technological microbes, which are key players in shaping the cheese organoleptic characteristics [46]. Therefore, pasteurization and low pH contribute to the selection of specific bacteria in all types of cheeses, including some artisanal cheeses made without the use of starter cultures, such as fresh acid-set cheeses that have been curdled by the addition of acid (e.g., Mascarpone, Queso blanco, Paneer). On the other hand, the use of raw milk for “traditional cheese” label production, inoculated or not with starter cultures, results in a richer ripening microbiota that shapes distinctive cheese sensory properties [47,48].

As mentioned before, the type of milk and its indigenous microorganisms play an essential role in shaping cheese microbiota and thus, cheese sensory characteristics [49]. Moreover, manufacturing conditions, such as the type of rennet, the type of starters and adjuncts used, salting, scalding temperature, smoking, as well as ripening conditions, such as temperature and humidity, are significant factors as well [33,50,51,52,53,54].

### 2.3. The Cheese Microbial Pool

The cheese microbiome consisting of bacteria and yeasts/fungi is developed during the two essential manufacturing steps, namely fermentation and ripening. The bacterial pool includes:(a)The dominant population of the pro-technological SLAB used either individually or in various combinations depending on the cheese variety, for their metabolic capacity to successfully ferment lactose to lactic acid, as well as their contribution to the development of cheese sensorial characteristics [12]. The back-slopping technique has been traditionally used for the production of artisanal cheese products, requiring the inoculation of milk with natural milk or whey cultures consisting of unknown strains, termed as undefined starters [55]. Nowadays, starters consist of a specific cocktail of well-defined strains [55], as their survival and spatial distribution within the cheese matrix are strain-dependent properties and can determine the final populations in the curd [56]. Mesophilic cultures, mainly strains of *Lactococcus lactis* (both subsp. *lactis* and *cremoris*), as well as thermophilic cultures, primarily *Streptococcus thermophilus*, are the commonly used starters. Additionally, members of the genus *Lactobacillus* are used as starter cultures, mainly *Lactobacillus delbrueckii* subsp. *bulgaricus* and subsp. *lactis* and *Lactobacillus helveticus*, and rarely *Lacticaseibacillus casei*, *Lacticaseibacillus paracasei*, *Lactiplantibacillus plantarum/paraplantarum*, *Levilactobacillus brevis*, and *Lacticaseibacillus rhamnosus* [55,57].(b)The NSLAB, originating from raw milk as well as the production and ripening environment, may exhibit proteolytic and lipolytic activities and metabolize other carbon sources than lactose, e.g., citrate, which can be catabolized by *Leuconostoc* spp., *L. casei*, *L. plantarum*, and *Weissella paramesenteroides*. NSLAB typically consist of facultative heterofermentative lactobacilli, including *L. casei*, *L. paracasei*, *L. rhamnosus*, *L. plantarum*, and *Latilactobacillus curvatus*, as well as obligate heterofermentative lactobacilli, including *L. brevis*, *Limosilactobacillus fermentum*, *Lactobacillus wasatchii* sp. nov., and *Lentilactobacillus parabuchneri* [55]. Other LAB, including species of the genera *Streptococcus*, e.g., *Streptococcus macedonicus* and *Streptococcus equi*, *Pediococcus*, *Weissella*, and *Enterococcus*, are part of the NSLAB group in many cheeses [58,59]. Enterococci, in some cases, have been used as experimental starter cultures; however, the genus *Enterococcus* does not possess Qualified Presumption of Safety (QPS) status, due to its potential to acquire genetic determinants of virulence and antibiotic resistance [60]. Some NSLAB strains can be selected and deliberately used as adjunct cultures and may dominate the cheese microbial ecosystem, enhance flavor development, and provide health benefits [11,61,62]. In addition, they help to achieve a balanced cheese flavor and may exhibit important metabolic activities, such as proteolytic, lipolytic, and esterolytic activity; amino acid degradation; and citrate catabolism [63]. Adjunct starters include strains of *Lactobacillus* spp. as well as the citrate fermenting *Leuconostoc mesenteroides* and *Leuconostoc pseudomesenteroides* [55]. Additionally, genera, such as *Corynebacterium, Arthrobacter,* and *Brevibacterium*, which are very important to produce smear cheeses, as well as *Propionibacterium*, which is a significant starter in Swiss-type cheeses, can be also found in cheese even at low levels [64].(c)The spoilage and pathogenic bacteria frequently detected in cheese. They derive either from milk or the processing environment, and when good hygiene practices during the entire cheese processing and storage are not employed, they can either lead to production losses due to spoilage or undermine the safety of the final product [31]. They include species of the genera *Clostridium*, *Staphylococcus*, *Salmonella*, *Listeria*, *Brucella*, *Shigella*, *Bacillus*, *Escherichia*, *Pseudomonas*, *Citrobacter*, *Klebsiella*, *Enterobacter*, *Psychrobacter*, *Proteus*, *Serratia*, and *Halomonas*, many of which are psychrotolerant [57,65].

As with bacteria, the fungal pool of cheese microbiota includes:(a)The starter cultures that are directly added to the milk or curd. Several cheese varieties are produced with filamentous fungi. They belong to two categories, namely the internally ripened cheeses, such as blue cheeses and Gamalost produced with *Penicillium roqueforti* and *Mucor* spp., respectively, in which fungal growth is visible both in the interior part and on the surface of the cheese, and the surface-ripened cheeses, such as the Camembert-type soft cheeses produced with *Penicillium camemberti*, in which fungal growth is visible only on the surface of the cheese [66]. Strains of *Mucor* spp., *Trichothecium roseum*, *Fusarium domesticum*, *Scopulariopsis* spp., *Sporendonema casei*, *Geotrichum candidum* (teleomorph *Galactomyces candidus*), *Kluyveromyces lactis*, and *Debaryomyces hansenii* are also used for the production of mold and smear-ripened cheeses [66,67].(b)The non-starter yeasts and molds (NSYMs), including species of the genera, such as *Issatchenkia*, *Aspergillus*, *Cochliobolus*, *Fusarium*, *Mucor*, *Cryptococcus*, *Pichia*, *Saccharomyces*, *Kluyveromyces*, *Debaryomyces*, *Candida*, *Trichosporon*, and *Yarrowia,* can be transferred from raw milk to cheese or may be contaminants from the dairy environment and influence cheese ripening [67]. While some of the NSYMs are known to be spoilers (e.g., *Cladosporium cladosporioides*, *Mucor racemosus*, *Penicillium commune*), there are others with an unknown or positive contribution to the cheese sensorial and physicochemical characteristics [67]. Moreover, strains of fungal genera, such as *Penicillium*, *Aspergillus*, and *Fusarium* spp., are potential producers of mycotoxins, such as aflatoxin M1, ochratoxin A, citrinin, roquefortin C, mycophenolic, and cyclopiazonic acid, which have been detected in cheeses at various levels, while certain yeast species, such as *Yarrowia lipolytica*, are able to produce biogenic amines [68,69].

The use of HTS in profiling cheese microbial diversity revolutionized the way we understand the cheese ecosystem and the dynamic changes in the microbial communities that take place during cheese ripening [18,57,70]. HTS contributed to unraveling the microbial diversity of natural whey cultures, which are applied as thermophilic LAB starters in Protected Designation of Origin (PDO) cheeses [57]; detected several novel bacterial genera in the cheese or cheese rind in various types of cheese varieties, such as *Prevotella*, *Arthrobacter*, *Yaniella*, *Nocardiopsis*, *Pseudoalteromonas*, *Facklamia*, *Vibrio*, and the species *Geobacillus toebii* and *Methylobacterium populi* [33,71,72]; and revealed the dominance of bacteria, such as *Psychrobacter*, in some cheese varieties [73]. Furthermore, the potential contribution of some unusual species, such as *Pseudoalteromonas haloplanktis* and *Psychrobacter immobilis*, has now been revealed by metagenomic profiling of three types of cheese rinds, while four fungal genera (among them *Aspergillus* and *Sporandonema*) have recently been reported as part of the sub-dominant cheese microbiota, revealing the accessibility of the diversity of low-abundant taxa by the powerful HTS technologies [72].

Recently, the combination of modern molecular biology tools with conventional culture-based techniques revealed the presence of 104 bacterial and 39 fungal genera in the microbial communities of 33 cheese rinds [74], while the characterization of new metagenomes from 55 kinds of cheese offered 328 metagenome-assembled genomes, including 47 putative new species that could influence taste or color [28]. Such results confirm the richness of the cheese microbial diversity and inspire the interest for its study ad infinitum.

## 3. Exploring the Cheese Microbiome

### 3.1. Culture-Dependent Tools

Exploration of the diverse cheese microbial community, which varies not only among cheese types but also within the cheese from the core to the surface, commenced using culture-dependent microbiological methods [22,74,75]. This approach includes bacterial growth and enumeration on selective growth media, isolation and identification at the genus and species level, and finally, characterization of biotypes at the intraspecific level. Many selective media have been developed so far for both bacteria and fungi, regardless of whether they are pro-technological or spoilage and pathogenic ones. One should bear in mind, however, that the media selectivity is not necessarily optimal, while at the same time, they do not always support the growth of viable but not culturable (VNC) microorganisms [76].

Classification of cheese microorganisms initially relied on the phenotype, e.g., morphological, biochemical, and physiological traits [77]. More advanced phenotypic methods are based on the whole-cell protein profile using either sodium dodecyl sulphate-polyacrylamide gel electrophoresis (SDS-PAGE) or matrix-assisted laser desorption ionization-time of flight mass spectrometry (MALDI-TOF MS), fatty acid analysis, and immunology-based methods, such as enzyme-linked immunosorbent assay (ELISA) or serological testing [78,79].

Genotypic techniques provide, however, wider and more reliable information concerning the diversity of the cheese microbiota at the genus, species, and even strain level; they, thus, enhance the potential for exploitation of the cheese microbial community in this biochemically dynamic product, as well as the development of innovative cheese products [77]. Outstanding bacteria identification methods include species-specific polymerase chain reaction (PCR), amplified ribosomal DNA restriction analysis (16S-ARDRA), sequencing of 16S ribosomal RNA gene (16S rRNA PCR), tRNAAla-23S rDNA-restriction fragment length polymorphism (tRNA Ala–23S rDNA-RFLP), sequencing of the gene encoding for phenylalanyl-tRNA synthase alpha-subunit (pheS), and multiplex real-time PCR (mRealT-PCR) using the pheS as a molecular target, while DNA fingerprinting techniques include RFLP of protein-coding genes involved in primary metabolism, restriction enzyme analysis pulsed-field gel electrophoresis (REA-PFGE), randomly amplified polymorphic DNA (RAPD), amplified fragment length polymorphism (AFLP), and repetitive extragenic palindromic-PCR (REP-PCR) [74,76]. Similar methods are used for the identification of fungi spanning the internal transcribed spacers (ITSs) and targeting the 5.8S, 18S, 26S, or 28S region of the nuclear ribosomal RNA gene [80,81,82,83,84,85].

### 3.2. Culture-Independent Tools

#### 3.2.1. Conventional Methods

The limits of the culture-dependent approach due to its inefficiency to detect “difficult-to-culture” or sub-dominant microorganisms have moved the trend toward culture-independent methods that avoid the use of selective cultivation and isolation [22,76]. The most common procedure for the culture-independent approach is the analysis of nucleic acids, either DNA or RNA, isolated from an entire microbial population using PCR amplification of the target sequences. The list of widely applied methods includes polymerase chain reaction denaturing gradient gel electrophoresis (PCR-DGGE), PCR-temporal temperature gradient gel electrophoresis (PCR-TTGE), single-strand conformation polymorphism-PCR (SSCP-PCR), terminal-RFLP (T-RFLP), length heterogeneity-PCR (LH-PCR) analysis, quantitative real-time PCR (qPCR), and reverse transcription-qPCR (RT-qPCR), as well as denaturing high-performance liquid chromatography (DHPLC) and fluorescence in situ hybridization (FISH), which do not rely on PCR amplification [76,86,87]. The 16S and 26S rRNA-encoding genes are the most frequently employed targets for identifying bacterial and eukaryote species, respectively [88,89], although other genes, such as *pheS* and RNA polymerase B subunit (*rpoB*), have also been used as targets for bacterial identification in cheese [90,91].

#### 3.2.2. Novel Omics Approaches

Given the complexity of the cheese microbiota as well as its plasticity, i.e., the dynamic succession of microbial groups during cheese making and ripening, the major challenge is to reliably assess the route and fate of each microbial group or even certain microbial strains during cheese making, and to understand their role and contribution in the quality and safety of the final product. The above challenges surpass the limits of the established classical and advanced microbiological and chemical analyses and can nowadays be overcome by the application of the so-called -omics approaches.

The term-omics refers to their ability to characterize in a single analysis all or most members of a family of molecules involved in the development and maintenance of life in a non-targeted and non-biased manner [92] and they include high-throughput next-generation sequencing (NGS) -based methods [93], such as genomics/metagenomics and metatranscriptomics targeting DNA and RNA, respectively. Additionally, metaproteomics and metabolomics, targeting proteins and metabolites, respectively, are of high significance as well (Figure 1). Their application to cheese microbiota will be discussed in the following sections.

## 4. Cheese Microbiota Metabolic Pathways for Flavor Development

Cheese flavor development is a dynamic process that is influenced by the type and composition of milk, processing conditions, and biochemical reactions of microorganisms present in the cheese matrix [94]. Besides rennet and milk indigenous enzymes, cheese microbiota is a primary source of enzymes participating in biochemical pathways involved in cheese manufacture and ripening; hence, it plays a key role in shaping the sensorial properties of the final product. In fact, population interactions among both bacteria and fungi (yeasts and molds) lead to a variety of flavor compounds present in the different types of cheese contributing to differentiate them [95]. Substrates for biochemical processes are carbohydrates, lipids, and proteins, with the latter being considered as the principal ones for cheese flavor formation [55].

### 4.1. Carbohydrate Catabolism

The catabolism of the principal milk carbohydrate lactose is the first fermentation step, upon which all milk-based fermentation processes are based [20]. The pathways through which lactose is catabolized depend considerably on the starter cultures, as well as the type and abundance of NSLAB for each cheese variety [96]. Starter activity, curd washing, and cheese salting impact the amount of residual lactose available to bacteria after cheese manufacture as most of the lactose is lost in the whey. Initial catabolism of lactose, via the Embden–Meyerhof–Parnas (EMP) pathway and/or the Leloir pathway after uptake and cleavage to glucose and galactose moieties, gives a mixture of L- and D-lactate depending on the starter cultures [96,97]. Moreover, the tagatose-6P pathway is also involved in the catabolism of galactose-6P, with end products also entering the EMP pathway (Figure 2) [98]. Residual lactose in the curd is rapidly depleted, mainly by NSLAB, during the early stages of ripening [99]. Heterofermentative NSLAB bacteria, such as *Leuconostoc* and certain *Lactobacillus* species, produce, besides lactate, ethanol, acetate, and CO_2_ via the phosphoketolase (PK) pathway [100]. Furthermore, *Leuconostoc* spp. along with *Lactococcus lactis* subsp. *lactis* biovar. *diacetylactis*, can convert pyruvate, an intermediate molecule in several metabolic pathways, to diacetyl and acetaldehyde, products commonly found in Gouda and fresh milk cheeses. Finally, lactate, derived from the action of both starters and NSLAB, becomes the substrate for a range of biochemical reactions that lead to the production of flavor compounds. For instance, *Propionibacterium* spp. can convert lactate to flavor-forming compounds, such as propionate and acetate that contribute to the Swiss-type cheese flavor. Regarding yeasts, lactate catabolism leads to the production of CO_2_ and ethanol, which have been correlated to non-desirable off-flavors of cheeses [101]. Moreover, lactose-fermenting yeasts, namely *K. lactis*, *Kluyveromyces marxianus*, and *D. hansenii*, that are present in certain cheese varieties are able to assimilate lactose, leading to the accumulation of ethanol and acetic acid [102].

### 4.2. Citrate Catabolism

Interestingly, some LAB also utilize citrate in the absence of carbohydrates. Citrate, although present at relatively low concentrations in milk (approximately 10 mM), can have a profound impact on cheese aroma. Only few LAB species, such as *L. lactis* and *Leuc. mesenteroides*, are able to utilize citrate and this property is linked to plasmid-encoded citrate permease genes (*citP*) regulated under low pH conditions in the absence of sugars [97,105]. Once inside the cell, citrate is cleaved into acetate and oxaloacetate by the enzyme citrate lyase, and then, oxaloacetate is converted into pyruvate and CO_2_ by oxaloacetate decarboxylase [103] (Figure 3). Although carbon dioxide is responsible for cavity formation in certain cheese types, regarding flavor development, the co-metabolism of citrate and lactose leads to characteristic C4 aroma compounds, such as diacetyl, acetoin, and 2,3-butanediol. In detail, bacterial α-acetolactate synthase catalyzes the condensation of two pyruvate molecules to give α-acetolactate, especially under conditions of pyruvate excess and acidic pH. Once synthesized, α-acetolactate is unstable and is either decarboxylated to acetoin by α-acetolactate decarboxylases or to diacetyl by non-enzymatic decarboxylation (in the presence of oxygen, though). Acetoin can also be synthesized from diacetyl by diacetyl-acetoin reductase, an enzyme that converts acetoin to 2,3-butanediol [106].

### 4.3. Lipid Catabolism

Cheese, in general, has a high fat content (even reaching 35% *w*/*w* in certain varieties) [107] and lipids may undergo oxidative or hydrolytic degradation. As lipid oxidation does not occur at a significant extent in cheese due to its low redox potential and the presence of natural and synthetic antioxidants, its contribution to flavor formation is limited [108]. However, enzymatic hydrolysis, especially during cheese ripening, has a major contribution to flavor development in many cheese varieties [109].

Raw milk contains a potent indigenous lipoprotein lipase (LPL), unstable to heat, which generally causes low levels of lipolysis in cheeses made from pasteurized milk, but its action is significant in raw milk cheeses [110]. Consequently, it is the cheese microbiota that contributes more than endogenous enzymes to flavor formation in cheeses produced from pasteurized milk. In general, most lipolytic enzymes are specific for fatty acids (FAs) esterified at the sn − 1 or sn − 3 positions of milk triglycerides, such as butyric acid and other short- and medium-chain FAs [111]. There are two types of lipolytic enzymes, namely lipases that are active on lipids, and esterases that are active on water-soluble ester substrates [111].

LAB are generally considered to be weakly lipolytic; however, their activity is substantially higher on diglycerides and monoglycerides, especially when the esterified fatty acid is a short-chain fatty acid (SCFA) [97]. Esterase activities have been detected in streptococci, lactococci, and mesophilic and thermophilic lactobacilli [111]. SCFAs can directly contribute to flavor, but the actual flavor formation follows, as free fatty acids (FFAs) can act as precursors for the production of a wide range of flavor compounds [6]. Specifically, esters can be formed in cheese via esterification (reaction of FFAs and alcohols) and alcoholysis (transfer of an acyl group from glycerides to an alcohol), with the involved pathways being species and strain dependent [112]. The presence of ethanol as the most abundant alcohol in cheese explains the prevalence of ethyl esters, compounds associated with fruity notes in cheese [111]. Other enzymatic reactions using FFAs as substrates are β-oxidation and decarboxylation and the produced methyl ketones and secondary alcohols play a major role in cheese flavor. Additionally, esterification of hydroxy-fatty acids produces lactones that also contribute to cheese flavor [113] (Figure 4).

Microorganisms other than LAB significantly contribute to flavor development through lipolysis, for many cheese varieties. Propionic acid bacteria (PAB) are well known for their lipolytic activity, which was found to be 10–100 times higher than the one attributed to LAB. For instance, *Propionibacterium freudenreichii* releases FFAs during the ripening of Swiss-type cheeses and lipolysis occurs with an initial preferential release of butyric acid followed by palmitic acid [114]. Regarding mold-ripened cheeses, fungi possess significant lipid degradation ability contributing to flavor formation [20]. For instance, *P. roqueforti* produces extracellular lipases, which are responsible for the extensive lipolysis of Roquefort cheese, while *P. camemberti*, along with the complex bacterial surface microbiota, produce lipases that contribute to the characteristic flavor development in white mold-ripened cheese varieties [96].

### 4.4. Protein and Amino Acid Catabolism

Proteolysis can be considered the key biochemical event in cheese flavor formation, leading to the release of peptides and amino acids that can be further catabolized to various flavor compounds. Proteolysis in cheese starts in the coagulant with enzymes (e.g., rennin), resulting in the formation of large- or intermediate-sized peptides. With the major part of rennin being lost in the whey during curd drainage, LAB play a significant role in milk proteins breakdown, in particular during cheese ripening, despite their weak proteolytic system. Generally, SLAB have a superior role in protein/peptide breakdown than NSLAB [11]. The LAB proteolytic system comprises cell envelope-associated extracellular proteinases; cell membrane transport systems for oligopeptides, di-/tripeptides, and free amino acids (FAAs); and intracellular peptidases [115]. FAAs are key substrates for a variety of biochemical reactions, leading to the production of major flavor compounds identified in many cheese varieties [116] (Figure 5).

Conversion of branched-chain amino acids (leucine, isoleucine, and valine) and aromatic amino acids (tryptophan, phenylalanine, and tyrosine) to the respective α-keto acids begins with a transamination step, and the aforementioned α-keto acids can then be converted into their corresponding aldehydes, carboxylic acids, alcohols, and their related (thio)esters [7]. Sulfur-containing amino acid catabolism (methionine and cysteine) leads to the formation of methanethiol (MTL) and other sulfur derivatives. Specifically, methionine catabolic pathways leading to MTL vary among LAB, with two enzymatic pathways being suggested: the most direct pathway involves C–S lyases (cystathionine β- or γ-lyases), while the other one involves the formation of α-keto-γ-methylthiobutyric acid (KMBA) that is further converted to MTL [117]. MTL, as well as dimethyl sulfide, dimethyl disulfide, and dimethyl trisulfide, all three obtained after MTL auto-oxidization, are regarded as essential cheese flavor components [103].

Regarding yeasts, it is generally recognized that the species *G. candidum*, *Yersinia lipolytica*, and *K. marxianus* are more proteolytic than *D. hansenii*. In fact, yeasts are able to produce volatile compounds that contribute to cheese flavor, such as branched-chain aldehydes and alcohols [15]. Moreover, fungi belonging to the genus *Penicillium* are responsible for intense proteolytic activity during cheese maturation in mold-ripened varieties, such as Camembert and Brie, with the released ammonium compounds playing a vital role in cheese flavor development [118].

## 5. Omics Insights into Flavor Formation in Cheese

### 5.1. Genomics

The first omics technique to appear, genomics, focused on sequencing and annotation of the complete genome of an organism in order to reveal the genetic structure and to predict gene functions that are encoded within the genome. Since the publication of the long-awaited whole genome sequence of the first LAB strain in 2001, namely *L. lactis* subsp. *lactis* IL1403 [119], the number of sequenced LAB genomes has grown exponentially, due to the advent of HTS, with the genome sequences of thousands of LAB species and strains currently being deposited in public databases [120].

The study of Bolotin et al. [119] was the first genomic milestone in LAB research and laid the groundwork for a better understanding of the many aspects of bacterial physiology, metabolic pathways, and regulatory mechanisms, as well as how phenotypes are affected by genetic variations. In the following years, numerous genomics studies were conducted regarding strains of the “laboratory workhorse” species *L. lactis* [121,122,123,124,125]. As mentioned before, *S. thermophilus*, *L. bulgaricus,* and *L. helveticus* are also the main SLAB used in cheese production along with *L. lactis*. Therefore, due to their industrial and economic importance, efforts are constantly being made to fully sequence and annotate genomes of these taxa [126,127,128,129]. In recent years though, genomics studies included NSLAB as well, due to their considerable impact on flavor development in cheese [98,130,131,132,133,134,135,136,137,138,139,140].

#### Proteins Associated with Flavor Formation in Cheese

Proteolysis, lipolysis, and AAs/FAs catabolism have all been studied in detail before whole genome sequencing (WGS) appeared. However, the available genome sequences allowed an in-depth analysis of the respective genes, so as to understand the genetic instability of several traits and unravel strain-specific differences. From this perspective, a comparative genomics analysis of all the proteins involved in the proteolytic system of 22 completely sequenced LAB strains was performed, including the cell-wall-bound proteinase, peptide transporters, and peptidases [115]. Based on the results, the genomes of the *Lactobacillus acidophilus*, *Lactobacillus johnsonii*, *Lactobacillus gasseri*, *L. bulgaricus*, and *L. helveticus* strains studied encoded a relatively higher number of proteolysis-related genes. Furthermore, the cell-wall-bound proteinase PrtP was solely identified in the chromosomes of *L. acidophilus*, *L. johnsonii*, *L. bulgaricus*, *L. casei*, *L. rhamnosus*, and *S. thermophilus* strains, as well as in the plasmid of *L. lactis* subsp. *cremoris* SK11. On the other hand, endopeptidases PepE/PepG and proline peptidases PepI, PepR, and PepL were absent in *Lactococcus* and *Streptococcus* strains, while aminopeptidases PepC, PepN, and PepM and proline peptidases PepX and PepQ were present in all genomes analyzed [115]. More recently, comparative genomics analysis among 213 assemblies, of which 175 belonged to *Lactobacillus* species and 38 to associated genera, was performed [141]. Regarding the metabolic potential of the 213 strains analyzed, the authors found genes for 60 cell envelope proteinases, which are important for cleaving casein during growth in milk and thus, contribute to cheese flavor, ranging in length from 1097 to 2270 amino acids. In addition, a broad repertoire of glycoside hydrolases and glycosyltransferases was identified, which are both important in carbohydrate metabolism [141]. Therefore, genome-scale metabolic models have been constructed and applied for the in silico prediction of the metabolic patterns of LAB strains under various conditions [142,143,144,145,146,147]. It should be noted, however, that the accuracy of these models depends on the quality of the genome sequencing and the correct annotation.

Furthermore, genomes of the non-LAB genera *Propionibacterium* and *Brevibacterium* have also been sequenced and annotated, as they also contribute to the flavor of certain types of cheeses. Genomics analysis of *Propionibacterium* assemblies mainly focuses on the identification of genes that are involved in the two key metabolic pathways for the propionate production, i.e., the Wood–Werkman and the tricarboxylic acid cycles, the amino acid catabolic pathways, which result in the formation of volatile compounds, and the detection of esterases involved in the formation of FFAs and esters [148,149,150]. Similarly, WGS studies of *Brevibacterium* assemblies reported, among others, the gene repertoire responsible for the catabolism of lactose, galactose, citrate, lipids, proteins, and amino acids, which contribute to the flavor, texture, and appearance of cheeses [151,152,153,154].

Apart from bacteria, several fungal species are also used as starters in internally and surface-ripened cheeses, such as Roquefort and Camembert, respectively, with a vital role in the flavor formation of the final product. Albeit their importance, there is still a limited number of sequenced genomes in the NCBI database, e.g., only five and six partially sequenced assemblies for *P. roqueforti* and *P. camemberti*, respectively, which are two of the main fungal species used in cheese production. However, most of the WGS studies performed focused on the phylogeny and not the metabolic pathways regarding flavor formation [155,156,157,158,159].

Linking genotype to phenotype is of paramount importance to an in-depth understanding of the technological potential of a strain and therefore, to a better selection of candidates with flavor-forming metabolic potential to be used as starter or adjunct cultures in cheese production. However, as cheese ripening depends on a complex microbial community, the need for metagenomics analysis quickly arose.

### 5.2. Metagenomics

Metagenomics encompasses two different HTS approaches, namely amplicon sequencing and shotgun metagenome sequencing. In amplicon sequencing, a highly conserved marker gene or genome’s region of DNA directly extracted from a food microbial community, e.g., that of cheese, is amplified and sequenced. These markers are of taxonomic relevance, with the 16S rRNA gene being used for the identification of bacteria taxa and the 18S rRNA gene together with the ITS DNA region for the yeasts/fungal taxa. On the other hand, in shotgun metagenomics, the extracted DNA is fully sequenced in a non-targeted manner, thus providing not only taxonomical identification results, but also information on the metabolic potential of the microbial community by reconstructing metabolic pathways [19].

From this perspective, a shotgun metagenomics study was recently performed in 25 Cotija cheese samples [160]. This artisanal Mexican cheese is produced by raw milk without the addition of starter cultures and is ripened in an open environment. Therefore, the organoleptic characteristics of Cotija cheese depend on a range of biotic and abiotic factors. Taxonomic identification revealed that the bacterial microbiota is mainly composed of Firmicutes, followed by Actinobacteria and Proteobacteria. The authors were able to reconstruct the genome assemblies of the three dominant species detected, i.e., *L. plantarum, Leuc. mesenteroides,* and *W. paramesenteroides*, which accounted for more than 80% of the total bacterial sequences. Gene functional annotation related to Cotija cheese flavor resulted in the identification of genes involved in the catabolism of phenylalanine, branched-chain amino acids, and fatty acids [160].

### 5.3. Transcriptomics

Genomics is widely applied to study the technological potential of a strain; however, it can only in silico predict its functional potential. Therefore, transcriptomics (RNA-seq), although a relatively new field of research, has gained much attention as it provides an accurate profile of a strain’s functional activity at a given time point.

Regarding cheese microbiota, the majority of the transcriptomic studies have been performed on *L. lactis*, due to its importance in cheese production [120,161,162,163,164,165,166,167,168,169]. In particular, the transcriptional responses of *L. lactis* have been investigated during different stresses, such as cold, heat, acid, osmotic, oxidative, and starvation [170,171,172,173,174,175,176], or during growth in media with different carbon sources [177,178,179]. Apart from *L. lactis*, transcriptomics studies have also been performed in other important cheese bacterial [163,180,181,182,183,184,185,186], yeasts [187,188,189], and fungal species [190,191,192,193,194,195,196]. Most interestingly, Dalmasso et al. [197] reported that *P. freudenreichii* CIRM-BIA1^T^ adapted from warm (28 °C) to cold storage (4 °C) was able to express genes involved in the formation of important cheese flavor compounds through the catabolism of branched-chain amino acids. Similarly, the transcriptome profile of *P. freudenreichii* CIRM-BIA138 was analyzed during the adaptation of the strain to starvation [198], while that of *P. freudenreichii* ITG P14 during different stresses, i.e., cold, heat, and starvation [199]. Moreover, the transcriptome profile of *L. rhamnosus* PR1019 was recently evaluated in a cheese-like medium during carbon source starvation [200]. The analysis revealed that the strain was able to adapt under these conditions using alternative metabolic pathways, such as pyruvate degradation and ribose catabolism.

### 5.4. Metatranscriptomics

In contrast to the plethora of transcriptomics studies in single microbes, a limited number of metatranscriptomics projects have been performed to assess the entire gene expression of a food microbial community, such as that of cheese. Among the key problems are the short half-life of mRNA, the large amount of mRNA needed for analysis, the experimental design, and, most importantly, the selection of the proper sampling points and some technology-specific limitations, including the available bioinformatics tools and workflows for data processing and analysis [201]. The first comprehensive metatranscriptomics study was performed by Lessard et al. [202]. The authors monitored the metatranscriptome profiles of *G. candidum* and *P. camemberti* in the rind of an industrial Canadian Camembert-type cheese. Based on the functional annotation performed, transcripts related to energy metabolism, such as glycolysis/gluconeogenesis, the pentose phosphate pathway, the tricarboxylic acid cycle, and oxidative phosphorylation, were identified. Furthermore, lyases involved in the production of volatile sulfur compounds during methionine catabolism as well as transcripts related to cabbage sulfur aroma development and ammonia production were also detected [202].

More recently, Monnet et al. [203] evaluated the functional activity of the microbiome in the rind of a French Reblochon-type cheese, which was produced by *S. thermophilus, L. bulgaricus, G. candidum*, and *D. hansenii*. Metatranscriptomics analysis revealed that during ripening, only minor changes occurred in the LAB, such as an upregulation of genes involved in lipid and carbohydrate metabolism, while in yeasts, a significant upregulation of genes related to amino acid catabolism occurred from day 14 to day 35 (end of ripening), suggesting their contribution to flavor formation [203].

However, discrepancies observed between mRNA levels and protein abundance, which have been attributed to the variable half-lives of mRNA, variable rates of protein synthesis, and possible post-translational modifications of proteins [18], make the analysis of expressed proteins a valuable tool for investigating the functional activity of microbiota.

### 5.5. Metaproteomics

Metaproteomics is the large-scale characterization of the entire protein complement of microbial communities in a biological system of increased complexity at a given time point [204]. The progress of proteomics has been driven by the development of new technologies for peptide/protein separation, isotope labeling for quantification, and bioinformatics data analysis, while the analysis and quantification of proteins has been revolutionized by MS-based methods [205]. There are several challenges that must be overcome to address a metaproteomic study in animal food products. Among them are the presence of raw material proteins that interfere with the detection of microbial proteins, the complexity of the microbiota-expressed proteins in a fermented food present at concentration ranges that may vary dramatically, and finally, the presence of certain highly abundant proteins that are often not interesting for metaproteomic analysis, while the microbiota proteins may be several orders of magnitude less abundant. For these reasons, large-scale metaproteomic studies are limited. However, they are gradually gaining attention in the field of fermented food research, including cheese [206,207], to assess the functional diversity of microbial communities. To our knowledge, there are no metaproteomics studies on cheese flavor formation and development. However, the proteomic approach applied in a survey of bacterial proteins released in Emmental cheese revealed functional groups of proteins involved in proteolysis and glycolysis, pathways that influence the organoleptic characteristics of cheese [208].

### 5.6. Metabolomics

#### 5.6.1. Analytical Techniques

Metabolomics is a relatively new field of omics research dealing with the simultaneous and high-throughput identification and quantification of low molecular mass (<1500 Da) metabolites that are not genetically encoded and are produced and modified by the metabolism of living organisms, such as organic acids, carbohydrates, amino acids, peptides, nucleic acids, vitamins, polyphenols, alkaloids, and minerals [209]. Metabolomic analyses have been classified as targeted (specific) or untargeted (non-selective) analyses.

The vast chemical diversity of the compounds in the metabolome requires efficient metabolite extraction, chromatographic separation, mass spectral detection, identification, quantification, and multivariate data analysis. Various analytical techniques are used for separation, such as liquid chromatography (LC), including high-performance LC (HPLC) or ultra-performance LC (UPLC), gas chromatography (GC), and capillary electrophoresis (CE), coupled to mass spectrometry (MS). MS-based platforms predominate because of both their ability to identify a wide range of compounds and their high throughput capacity. Apart from MS, other commonly used detection techniques are nuclear magnetic resonance (NMR) and near infrared spectrometry (NIR). However, none of the individual analytical methods are capable of effectively analyzing all the metabolites of a sample.

#### 5.6.2. Metabolomics to Assess Flavor Formation in Cheese

Metabolomic studies in cheese provide insights on specific metabolites produced by cheese microbiota, with special interest in flavor compounds, and have significantly enabled characterization of the metabolite diversity in cheese, as well as the factors affecting it. According to Ochi et al. [210], GC/TOF-MS fingerprinting of hydrophilic low-molecular-mass compounds can be the basis of prediction models for sensory attributes, such as “rich flavor” and “sour flavor” in ripened Cheddar and Gouda, while specific amino acids [211] or volatile compounds, such as hexanoic acid, heptanoic acid, octanoic acid, 2-decenal, and acetoin [212], may be used as ripening markers in Cheddar cheese.

The cheese metabolome is affected by the degree of LAB autolysis. Grana-Padano cheeses produced in two different dairies and with different volatile profiles, as shown by solid phase micro-extraction (SPME) GC-MS, were analyzed with respect to their total culturable lactic microbiota and starter lysis. Remarkably, it was shown that increased complexity of microbial origin volatiles, such as ketones, alcohols, hydrocarbons, acetic acid, and propionic acid, was associated with the complex microbiota composition, with NSLAB (mainly *L. rhamnosus*/*L. casei*) being dominant. On the other hand, intracellular enzymes released due to SLAB cell lysis were involved in a higher content of FFAs, benzaldehyde, and organic acids, such as pyroglutamic and citric acid [213].

LC high-resolution MS (LC–HRMS) metabolic fingerprinting in model cheese revealed that the cheese metabolome is influenced by the spatial distribution of the starter *L. lactis* colonies throughout ripening [214]. By varying the time of renneting (at 0 h, e.g., simultaneously, or after 8 h from starter inoculation), they generated two different spatial distributions of immobilized bacterial colonies (few big colonies spread away from each other, or numerous small colonies close to each other, respectively), and identified 26 metabolites, including amino acids, organic acids, vitamins, nucleotides, and proteolysis products, being more abundant in small-colony cheeses. Moreover, according to Le Boucher et al., bacterial cells forming small colonies use the same metabolic pathways but display higher metabolic activity than big colonies, resulting in higher concentrations of metabolites being accumulated due to proteolysis or carbohydrate catabolism [215].

Investigation of the flavor compounds produced by the whole cheese microbiota or the individual contribution of SLAB or NSLAB strains is of great interest for cheese producers. Studies of this type are often performed in a cheese-based medium, mimicking the cheese environment, thus providing the nutrients and precursors present in cheese during ripening.

Sgarbi et al. explored the volatile compounds produced by different *L. casei* and *L. rhamnosus* strains grown on either a cheese-based medium (CBM) or on a LAB cell lysate-based medium (LCM). Volatile analysis by SPME GC-MS showed differences between NSLAB strains grown in CBM (where pyruvate could be a common precursor for all compounds produced) and LCM (where in the absence of lactate and citrate, NSLAB strains used mainly FFAs and FAAs), thus providing a better understanding of how NSLAB-produced volatile flavor compounds contribute to the development of cheese flavor during ripening, a first step toward the selection of wild NSLAB possessing a specific aromatic profile, for use as adjunct culture [216].

Similarly, Pogačić et al. evaluated the potential of *Lactobacillus* spp. and *Leuconostoc* spp. in the production of aroma compounds by incubating single strains in a curd-based slurry medium [217]. Analysis of volatiles by GC–MS revealed strain to strain variation. Acetoin, diacetyl, acids, and esters were mainly produced by *L. rhamnosus* and *L. paracasei*, while *Leuconostoc* spp. were major producers of alcohols and esters. The same curd-based slurry medium has been used by Pogačić et al. to screen LAB, Actinobacteria, *P. freudenreichii*, and *Hafnia alvei* strains for their ability to produce aroma compounds. Forty-nine out of 52 aroma compounds identified differed in their abundance among the bacteria [218].

A cheese-based medium was also used by Guarrasi et al. to grow SLAB and NSLAB strains isolated from Caciocavallo Palermitano cheese to assess their contribution to the volatile organic compounds (VOCs) production in ripened cheese [219]. By applying head space (HS)-SPME GC-MS to the fermented substrates, they found that strains of *L. delbrueckii*, *L. casei*, *L. paracasei*, *L. rhamnosus*, and *Enterococcus gallinarum* mainly influenced the development of the characteristic ripened-cheese aromatic compounds.

Furthermore, according to Yee et al., strains of dairy propionibacteria exhibited large inter-species and intra-species diversity in their ability to produce different aroma compounds when grown in a cheese curd-based medium [220]. GC-MS data differentiated *P. freudenreichii* strains from each other, as well as from *Propionibacterium acidipropionici* strains. For the same compound, differences between strains of the same species were as high as ~500-fold, with *P. freudenreichii* strains harboring the widest potential to produce cheese aroma compounds, making it a potential candidate to modulate cheese flavor.

Except cheese-based medium, cheese model systems are important tools that facilitate the study of the impact of microbial metabolism on cheese sensorial characteristics formation since they can be prepared under controlled microbiological conditions, are more economical, reproducible, and easier to obtain.

Miniature cheeses were prepared by Ruggirello et al. [221] to evaluate *L. lactis* commercial starter cultures’ viability and ability to produce aroma compounds during ripening. Culture-independent and -dependent approaches, as well as HS-SPME GC-MS, revealed the persistence of *L. lactis* in cheese throughout cheese making and ripening, although in a metabolically active hypothetical viable but not culturable (VBNC) state, since the expression of *L. lactis* cystathionine β-lyase (*metC*) and α-acetolactate synthase (*als*) genes, involved in the biosynthesis of sulfur aroma compounds and diacetyl/acetoin, respectively, was partially associated with acetoin, diacetyl, 2,3-butanediol, and dimethyl disulfide production in ripened cheeses.

Irlinger et al. [222] applied GC-MS for metabolic profiling of the Gram-negative bacteria *Psychrobacter celer* and *H. alvei* when grown on a mini smear cheese. Both bacteria were able to colonize the cheese surface and compete in the microbial community, modifying at the same time the aromatic content of cheeses, highlighting the fact that less abundant microorganisms can have a significant impact on cheese flavor.

In a recent study, Suzuki-Iwashima et al. [223] investigated the combined effects of LAB starters and the white fungus *P. camemberti* on the production of volatile compounds during the ripening of a model white mold surface-ripened cheese. Metabolomics analysis by GC-MS showed that the early ripening period was characterized by metabolites, such as lactose, galactose, lactic acid, ethanol, diacetyl, acetoin, ethyl acetate, and sulfur compounds (dimethyl sulfide and dimethyl disulfide), that derived from carbohydrate metabolism of LAB, while fungal metabolism of proteins (with amino acid-derived compounds including branched aldehydes, such as 3-methyl butanal), and of fatty acids and proteins (with methyl ketones, fatty acids and amino acids) characterized the intermediate- and the late-ripening stages, respectively.

#### 5.6.3. Metabolomics Shed Light on the Ability of Adjunct Cultures to Produce Flavor Compounds

Starter bacteria combined with adjunct LAB strains selected for desirable metabolic potential is a common tool used to control and accelerate cheese ripening [224], enhance and improve flavor intensity [62], especially in cheeses made with pasteurized milk [11], and accelerate flavor development [103]. Metabolomics approaches have been a valuable tool to assess the contribution of the potential adjuncts on cheese flavor development [18,55,225].

So far, several studies have been performed evaluating not only the contribution of potential adjunct cultures to cheese flavor development, but also their persistence, technological performance, together with interactions with the starter cultures in cheese making experiments. In a recent study, Stefanovic et al. [138], selected three *L. paracasei* strains based on their proteolytic enzyme activities and ability to produce flavor compounds in cheese model systems [226] to be used as adjunct cultures in Cheddar cheese manufacture. The authors found that the *L. paracasei* strains contributed to the development and diversification of flavor-related compounds in short-aged cheeses. Different adjunct cultures did not influence the gross cheese composition, nor primary or secondary proteolysis or lipolysis. However, cheese volatile analysis by GC-MS showed variation in long-chain aldehydes, acids, and esters that originated from the metabolism of FFAs, suggesting that starter lipolytic activity produced the primary metabolites, which were further metabolized by the adjuncts into flavor-contributing compounds.

Bancalari et al. [227] evaluated the use of a wild *L. paracasei* strain, selected for its ability to produce in vitro acetoin and diacetyl, as adjunct culture to enhance the flavor formation in Caciotta-type cheese. Indeed, the adjunct strain was able to develop in curd and cheese, producing higher amounts of volatile compounds and organic acids as monitored through SPME GC-MS, thus differentiating the experimental Caciotta with respect to the control cheese. Moreover, Belkheir et al. [228] prepared a Tetilla-type cheese using as adjunct cultures a high-diacetyl producer *L. plantarum* strain together with a peptidolytic *L. brevis* strain producing volatile sulfur compounds, to study the volatile profile and sensory characteristics of the model cheese and to compare it with the PDO Tetilla cheese. Volatile analysis with SPME GC-MS showed an increased abundance of acetic acid, hexanoic acid, ethyl butanoate, and ethyl hexanoate, as well as higher scores for flavor preference in the cheese made with the two adjuncts than in the control cheese, highlighting the fact that the use of selected adjunct strains would differentiate the cheese sensory properties. Interestingly, when *Kocuria varians* (*Micrococcaceae*) and *Y. lipolytica*, selected for their proteolytic and lipolytic activities, were used as adjunct cultures, for the manufacture of experimental Tetilla cheese from pasteurized cow’s milk, Centeno et al. [229] were able to recover the traditional flavor and sensory characteristics of raw-milk PDO Tetilla cheese. The volatile profiles of cheese manufactured with both adjuncts, detected by GC-MS, showed enhanced formation of fatty acids, esters, and sulfur compounds, thus the modifying the flavor profile of the experimental Tetilla cheese, which was considered very similar to good-quality artisanal raw-milk cheese.

In a potentially probiotic Caciotta cheese, industrially produced with autochthonous putative probiotic *Lactobacillus* and *Kluyveromyces* strains as adjunct cultures, Pisano et al. [230] investigated the adjunct’s influence on the cheese chemical and microbiological composition and sensory properties. Cheese metabolome characterization by means of ^1^H NMR spectroscopy (for amino acids, organic acids, and carbohydrates) together with HPLC-diode array detector/evaporative light scattering detector (DAD/ELSD) for the cholesterol, α-tocopherol, and fatty acid composition, highlighted significant variations in the cheese metabolome both in terms of the ripening time and strain combination, with *Kluyveromyces* and *Lactobacillus* strains surviving the manufacturing process and retaining their viability till the end of ripening, suggesting that Caciotta cheese can be used as a carrier for probiotic bacteria delivery.

#### 5.6.4. Metabolomics as a Means of Cheese Authentication

Effective and reliable analytical methods are of paramount importance to securing the authenticity of PDO cheeses, aiming to protect both the product value and consumers. To this end, metabolomics-based approaches represent a powerful method to discriminate fraudulent varieties of a given food product [57,231,232].

Pisano et al. [233] analyzed the polar metabolite profiles by GC-MS, together with the predominant cultivable microbiota from buffalo and cow Mozzarella, in order to discriminate them. PDO buffalo Mozzarella exhibited a higher microbial diversity together with less psychrotrophic bacteria, while cow Mozzarella showed the highest counts of *S. thermophilus*, originating from the commercial starter culture. Furthermore, the polar metabolites reflected differences in the production protocols and microbiota complexity of theses cheeses, suggesting that the polar metabolite profile can be a promising tool to characterize and verify the authenticity of Italian buffalo Mozzarella. Moreover, Rocchetti et al. [234] characterized low-molecular-mass metabolites based on ultra-high-pressure liquid chromatography coupled with quadrupole time-of-flight MS (UHPLC/QTOF-MS), aiming to reveal differences between genuine PDO and non-PDO Grana Padano cheeses. Amino acids, oligopeptides, and fatty acids were the biomarkers with the highest discriminatory power.

Another authenticity problem involves adulteration related to non-declared processing methods, such as in the case of Fiore Sardo (FS) cheese, where the use of raw ovine milk is mandatory. Caboni et al. studied the polar low-molecular-mass metabolites, by GC-MS, aiming to discriminate FS cheese produced from raw or thermized milk. FAAs and saccharides were the metabolites that mostly changed, suggesting the polar low-molecular-mass metabolites as a potential biomarker for detecting milk thermization in ovine PDO cheeses [235].

### 5.7. Integration

Studies combining multiple-omics approaches, integrating genomics, transcriptomics, together with metabolomics, that is, a systems biology approach, provide a detailed picture of the cheese microbiota dynamics, as well as information on potential microbial interactions and their contribution with respect to cheese sensorial characteristics development. An overview of the most comprehensive studies combining multiple omics approaches to study flavor development in cheese can be found in Appendix A. In this section, integrating omics studies, which have been performed not only in mature cheese, but also during cheese ripening, are discussed, to gain deeper insights on microbial succession and metabolite production.

#### 5.7.1. Amplicon Metagenomics-Shotgun Metagenomics

The first pioneering study incorporating different -omics techniques was performed by Wolfe et al. [72]. The authors analyzed the rind microbial communities of 137 cheese samples, including 61 natural cheeses that were left undisturbed during the aging process, 52 washed with 20% *w*/*w* NaCl, and 24 bloomy cheese rinds, from different geographic regions, milk types, and milk treatments. Using amplicon-based metagenomics analysis, only 14 bacterial and 10 fungal genera were identified at higher than 1% average abundance. Interestingly, the microbiota was found to be associated with the rind type (natural, washed, and bloomy) and moisture instead of the geographic origin. Additionally, shotgun metagenomics was performed to assess the functional potential of the microbiota based on the different rind types. In particular, shotgun data identified genes involved in several metabolic pathways associated with flavor formation, such as the catabolism of cysteine and methionine, which are known to contribute to the production of volatile sulfur compounds, and that of valine, leucine, and isoleucine, which provide sweaty and putrid aromas. It should be noted that the majority of these genes were detected in the washed rind cheeses. Furthermore, the halotolerant γ-proteobacteria genus *Pseudoalteromonas*, originally associated with marine environments, was found for the first time in cheese microbiota and, in particular, in the natural and bloomy cheese rinds. The shotgun metagenomics results identified a few cold-adapted enzymes produced by *Pseudoalteromonas* spp. that participate in lipolysis and proteolysis. Therefore, the presence of *Pseudoalteromonas,* as part of the cheese microbial community, could be considered beneficial, as it can contribute to the development of flavor compounds in cheeses during ripening and storage at low temperatures [72].

#### 5.7.2. Amplicon Metagenomics-Metabolomics-Metatranscriptomics

De Pasquale et al. [236] used Fiore Sardo, Pecorino Siciliano, and Pecorino Toscano cheeses as hard cheese model systems to study the spatial distribution of metabolically active microbiota and its effect on secondary proteolysis and VOC production, since these cheeses present a decreasing NaCl gradient from the surface to the center and an opposite moisture trend, properties that affect the microbiota distribution. By combining 16S rRNA gene pyrosequencing (targeting RNA) and Purge and Trap coupled with GC-MS (PT GC-MS), they found that in all cheese varieties, the poorest VOC profile was detected in the core region, due to the low oxygen availability, with high levels of alcohols originating from aldehydes and methyl-ketones reduction. Mesophilic lactobacilli (predominantly *L. plantarum*) positively correlated to alcohols, aldehydes, methyl and branched esters, and sulfur compounds. Thermophilic LAB in Pecorino Siciliano (including *L. delbrueckii* and *S. thermophilus*) positively correlated with the total concentration of FAAs, in different cheese regions, as well as with alcohols and related esters, while *Brevibacterium* sp. present on the surface of Pecorino Toscano correlated with alcohols, aldehydes, ketones, esters, and sulfur compounds.

Recently, Turri et al. [237] characterized the mature Historic Rebel (HR) cheese, an Italian heritage cheese produced from raw cow milk in the Alps. Microbiota diversity, assessed by 16S rRNA gene amplicon sequencing, revealed that the core microbiota comprising *Streptococcus*, *Lactobacillus*, *Lactococcus*, *Leuconostoc*, and *Pediococcus* genera correlated positively with the VOCs hexanal, 2-heptanal, 3-hydroxybutan-2-one (or acetoin), and ethanol respectively, as determined by SPME GC-MS. Moreover, a lipidomics approach was included by applying Dynamic Headspace (DHS) GC-MS to analyze the terpene fraction and the polyunsaturated fatty acids composition of HR cheese, parameters that were closely related with pasture vegetation and feeding, respectively, and contribute to the richness of cheese flavor.

In a sophisticated study [188], metabolite analysis by Ultra HPLC-MS (UHPLC-MS) and HPLC-UV in combination with dual RNA-seq analysis were applied in ripened lab-scale cheeses to provide insight into the metabolic interactions in a simple synthetic community composed of three species commonly used for the production of smear-ripened cheese. A strong mutualistic interaction between *Brevibacterium aurantiacum* and *H. alvei* was proposed, according to which, proteases and lipases secreted by *B. aurantiacum* liberate energy compounds from caseins and triglycerides that stimulate *H. alvei*, which in turn produces siderophore that increases iron availability for *B. aurantiacum*. Furthermore, the proteolytic activity of *B. aurantiacum* led to increased methionine catabolism in *H. alvei* producing methanethiol, a precursor for a wide variety of volatile sulfur compounds that contribute to cheese flavor.

#### 5.7.3. Shotgun Metagenomics-Metatranscriptomics

Recently, Duru et al. [49] and DeFilippis et al. [26] studied the effect of modifying the cheese ripening temperature on microbial community structure and function. More specifically, in a thorough study, the metagenome and metatranscriptome profiles of the semi-hard Swiss-type Maasdam cheese were studied during warm (20 °C) and cold (5 °C) room ripening, using *L. lactis* subsp. *lactis* and *L. lactis* subsp. *cremoris* as starter strains and as adjunct cultures strains of *P. freudenreichii* subsp. *shermanii*, *L. rhamnosus*, and *L. helveticus* [49]. The authors constructed four genomes (one genome per species) from the shotgun data to near completeness (higher than 97%), and based on the mean DNA read coverage, *L. lactis* was found to be the dominant species. Annotation of genome assemblies and pathways reconstruction identified genes required for FFAs biosynthesis in all genomes, proteolytic enzymes only in the LAB, and lipolytic enzymes in the genomes of *L. lactis, L. rhamnosus,* and *P. freudenreichii*. Furthermore, genes for valine catabolism were found in *L. lactis* and *P. freudenreichii*, while those for methionine and cysteine catabolism in *L. lactis, L. rhamnosus,* and *L. helveticus* genomes. Therefore, all species are important for the flavor formation of Maasdam cheese to a different extent. RNA-seq analysis confirmed the dominance of *L. lactis*, as more than 85% of the transcript reads mapped uniquely to this species. Moreover, metatranscriptomic data showed that *L. lactis* was metabolically active despite the ripening temperature. However, this was not the case for the other species, as genes related to the central metabolism were downregulated during cold room ripening, suggesting that fewer flavor compounds were produced. On the other hand, according to De Filippis et al. [26], elevation of the ripening temperature of Caciocavallo Silano cheese from standard (16 °C) to experimental (20 °C) temperatures directly affected microbiota diversity and metabolism. More specifically, 16S rRNA amplicon and shotgun metatranscriptome sequencing revealed an increased relative abundance of NSLAB in cheese ripened at higher temperatures, together with differential expression of 651 genes. Furthermore, overexpression of proteolysis, lipolysis, amino, and fatty acid catabolism-related genes at 20 °C correlated with increased production of cheese VOCs, as found by SPME GC-MS, and significantly increased the cheese maturation rate.

The functional potential of the microbiome in an experimental surface-ripened cheese was recently assessed by Dugat-Bony et al. [238]. Using reference genomes, the assembly of the shotgun metagenomics data resulted in the genome construction of the nine microbes used for the production of the cheese, i.e., six bacteria species (*L. lactis*, *Staphylococcus equorum*, *Corynebacterium casei*, *H. alvei*, *B. aurantiacum*, and *Arthrobacter arilaitensis*), and three yeast species (*D. hansenii*, *G. candidum*, and *K. lactis*). In addition to the shotgun metagenomics, the authors also employed metatranscriptomics to link metabolically related transcript reads to the microbiome at specific time points during cheese ripening. RNA-seq analysis revealed that enzymes involved in lactose fermentation were found to be expressed by *L. lactis* and *K. lactis*. Moreover, lactate degradation was attributed to *D. hansenii* and *G. candidum*, due to the high levels of lactate dehydrogenase transcripts detected in these species. Furthermore, although several transcript reads associated with protein and lipid metabolism were mapped to *L. lactis*, it was found that *G. candidum* was the main contributor to proteolysis and lipolysis. In addition, *G. candidum* was also found to be a key microbe regarding the catabolism of amino acids and thus, the production of flavor compounds throughout ripening. However, it should be noted that transcripts related to amino acid catabolism also mapped to the *L. lactis* genome, mainly at the early stage of ripening, as well as to the *C. casei* and *H. alvei* genomes at the end of ripening [238].

#### 5.7.4. Genomics-Amplicon Metagenomics-Shotgun Metagenomics-Metabolomics

Furthermore, Zheng et al. [29] studied the correlations between microbial dynamics and evolution and flavor production during ripening of Kazak artisanal cheese by amplicon sequencing (16S rRNA and ITS loci) for microbiota diversity together with SPME GC-MS for the analysis of neutral VOCs and volatile FFAs. A total of eight bacterial and seven fungal genera were identified across all time points of the Kazak cheese ripening process, the most abundant being *Lactobacillus* and *Streptococcus* together with *Kluyveromyces* and *Torulaspora*, respectively. Different bacteria and yeast genera were considered as functional core microbiota for producing amino acids, fatty acids, and volatiles. *Acetobacter*, *Lactococcus*, *Staphylococcus*, and *Bacillus* positively correlated with 2-nonanone, acetoin, and benzaldehyde; *Kluyveromyces* with butanoic acid; and ethyl ester, *Issatchenkia* and *Candida* with n-decanoic acid and hexanoic acid, respectively, while *Aspergillus* had a positive correlation with heptanal and n-decanoic acid. Various correlations with fatty acids were also assigned. Recently, in a similar study, Penland et al. [239] characterized the Pélardon cheese microbiota and VOCs throughout cheese making and ripening by 16S rRNA and ITS gene sequencing, and HS GC-MS analysis, respectively. *L. lactis*, the main acidifying bacterium, decreased during ripening. *Leuc. mesenteroides* and *G. candidum* correlated with amino acid catabolism at the early ripening stages, while *L. paracasei* and *Enterococcus faecalis* together with the fungi *P. commune* and *Scopulariopsis brevicaulis* dominated during prolonged ripening and were positively correlated to major volatile compounds responsible for the goaty and earthy Pélardon cheese aroma.

Moreover, Bertuzzi et al. [25] applied whole-metagenome shotgun sequencing to study rind microbiota succession and metabolic potential in surface-ripened cheeses and to associate it with volatile compounds detected with HS-SPME GC-MS. Over the course of ripening, correlation analysis between microbiome and volatile data revealed strong relationships between individual microorganisms and volatiles. *D. hansenii* correlated with the production of alcohols and carboxylic acids originating from FAA and FFA metabolism; *Brevibacterium linens*, *G. candidum*, and *Staphylococcus xylosus* with sulfur compounds and 2-methyl-1-butanol; *Corynebacterium variablile* with ketones; and *Glutamicibacter arilaitensis* with ketones, alcohols, and acids.

*Macrococcus caseolyticus* subsp. *caseolyticus* strains have been associated with the secondary microflora of Ragusano and Fontina cheeses and it has been suggested that they may have a positive impact on the cheese flavor profile [240]. According to Mazhar et al. [241], who studied *Macrococcus* strains of dairy and non-dairy origin, whole-genome sequencing and comparative genome analysis, further supported with enzymatic assays, revealed the strains’ limited ability to catabolize amino acids and consequently to produce amino acid-derived flavor compounds. Interestingly, lipase and high esterolytic activities were detected and correlated with diverse volatiles detected by GC-MS, since most of them were mainly associated with FFA metabolism. This type of study would help to identify strains potentially useful for further investigation, as adjuncts producing novel and distinct flavor profiles.

Moreover, multi-omics analyses have been applied to group artisanal and industrial Cheddar cheeses based on type and brand [242], differentiate Cheddar cheese based on age and brand [23], and compare cheese of varying quality [24]. More specifically, three omics datasets (16S rRNA amplicon sequencing, untargeted GC-MS, and LC-MS metabolomics) were analyzed to identify relationships between the cheese microbiota and metabolites of artisanal and industrial Cheddar cheeses. Metabolites with extensive diversity were detected in both artisanal and industrial cheeses, with many of them significantly associated with each cheese type and specific LAB genera, thus making the discrimination between industrial and artisanal Cheddar cheeses possible. Among them, the metabolites *O*-methoxycatechol *O*-sulphate and 3-hydroxy propanoic acid in artisanal cheese were reported for the first time in foods. The former originating from the metabolism of dietary phenolics [243] correlated positively with *Streptococcus*, while the latter compound with several industrial applications [244], positively correlated with *Lactobacillus* and *Streptococcus*. Furthermore, *Pediococcus*, present only in artisanal cheese, was correlated to 21 metabolites that may influence cheese flavor [242]. In a subsequent study, microbiota and metabolite profiles of industrial Cheddar cheeses of different ripening ages made by different manufacturers were analyzed [23]. Age-specific markers including numerous amino acids and carboxylic acids, such as malic acid, hydroxy-glutaric acid, citric acid, lauric acid, myristic acid, pentadecanoic acid, and hexadecenoic acid, were found to be positively associated with the ripening age. New significant associations existing between cheese microbiota and metabolites were described, such as the levels of phenylalanine correlating positively with the presence of *Thermus* sp., which may have originated from hot water sources in the factory and has been implicated with the pink discoloration in cheese [245], as well as the negative association between cheese cholesterol and *S. thermophilus* abundance, which confirms previous reports on the cholesterol-lowering activity of *S. thermophilus* strains in vitro [246]. Finally, Cheddar cheeses of different qualities made by the same manufacturer were investigated in an attempt to identify biomarkers (microbiota taxa and metabolites) that could discriminate, at the molecular level, Cheddar cheeses of different sensory qualities [24]. The metabolites with greatest discriminatory power included proline, histidine, isoleucine, and aspartic acid, present in greater amounts in the high-quality cheese samples, together with stearic acid and octadecanol, which were more abundant in the low-quality cheese samples. The key discriminatory taxa were *Streptococcus* (presumably *S. thermophilus*) and *Lactococcus* (*L. lactis*), which were found in a higher relative abundance in the high-quality and the low-quality cheese samples, respectively.

New insights into cheese microbiome were obtained from an extensive meta-analysis of cheese microbiomes and corresponding volatilomes, where 328 metagenome-assembled genomes from 184 cheese metagenomes were recovered including 47 putative novel species, the majority of which belonged to halophilic genera (such as *Psychrobacter* and *Halomonas*) or to genera associated with the rind (for example, *Brevibacterium*, *Corynebacterium*, and *Arthrobacter*) [28]. Metabolic modeling of their genomes predicted that they could influence cheese taste or color through the secretion of volatiles or pigment biosynthesis. Moreover, the integration of strain-level metagenomics with metabolomics indicated that variations in the abundancies of strains corresponded to differences in the volatilome. Except the detailed characterization of cheese microbiota, this study highlights the combination of strain-level metagenomics with metabolomics to correlate strain abundance with volatile levels, therefore evaluating the effect of specific strains on flavor, since different strains of the same species may produce different metabolites [247].

Indeed, the impact of strain diversity on cheese rind microbial dynamics and functional outputs has recently been documented by Niccum et al. [248]. The authors constructed several synthetic cheese rind communities by inoculating onto cheese curd agar distinct combinations of strains from the species *Staphylococcus equorum*, *B. auranticum*, and *Brachybacterium alimentarium*, with comparative genomics demonstrating the communities’ phylogenomic diversity and variable genome content. The initial identical community composition diverged over time and resulted in substantial differences in dominant community taxa, possibly due to strain-level variations, resulting in different interactions with other community members. Differing responses were observed upon communities’ exposure to abiotic (6% *w*/*w* salt) or biotic (addition of a *Penicillium* strain) perturbations. Furthermore, divergence in community composition also drives community functional diversity, since variations were observed in pigment production, as well as in the composition of volatile organic compounds detected by HSSE GC-MS across the communities.

## 6. Conclusions

The cheese microbiome is a dynamic ecosystem that develops and evolves during cheese manufacture and ripening and shapes the quality, organoleptic properties, and safety of the final product, factors that determine consumers’ preferences. Application of -omics techniques has greatly facilitated the study of microbiota dynamics and evolution during cheese ripening, together with its contribution to the organoleptic properties’ formation. Furthermore, integrating individual meta-omics approaches in combination with data integration analysis provided deeper insights into microbiota–metabolite interactions that influence cheese flavor and quality.

With genomics, the genetic structure of cheese microbiota is revealed, gene functions that are encoded within the genome are predicted, while WGS allows an in-depth analysis, among others, of the genes involved in glycolysis, proteolysis, lipolysis, and AAs/FAs catabolism, leading to exploitation of the metabolic diversity of cheese microbiota. Metagenomics analysis, i.e., amplicon and shotgun sequencing, provides not only taxonomical identification results, but also information on the metabolic potential of the microbial community. Furthermore, metatranscriptomics studies assess the entire gene expression of cheese microbiota, and the regulation of genes involved in lipid, carbohydrate, and AA catabolism, revealing their contribution to flavor formation. However, it is clear that genomic or metagenomic studies cannot fully account for the strains’ flavor potential, and integration with metabolomic-based approaches is essential. The information provided by this approach can be used not only to explore the flavor metabolites produced, but also to elucidate the role of LAB strains used for cheese production and evaluate NSLAB’s contribution to cheese flavor development.

Despite the many advantages associated with the use of -omics, there are still several limitations. However, considering the economic impact associated with the optimization of cheese production, it is certain that -omics studies will greatly contribute to cheese production standardization, diversification, and optimization.

## Figures and Tables

**Figure 1 foods-11-00188-f001:**
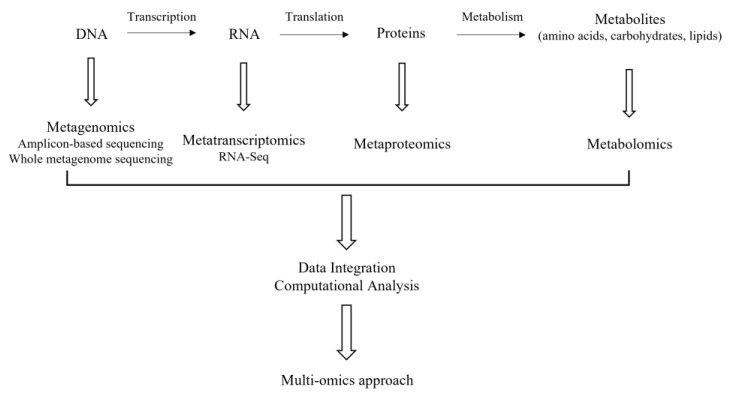
Integrating multi-omics approaches for the study of cheese flavor development.

**Figure 2 foods-11-00188-f002:**
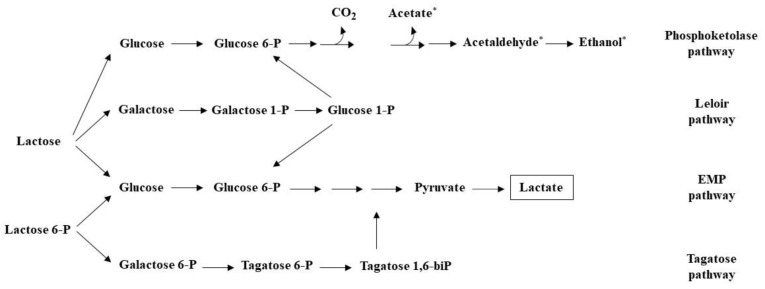
Schematic overview of carbohydrate catabolism pathways by LAB leading to the generation of flavor compounds in cheese. Asterisks denote flavor compounds [6,103,104].

**Figure 3 foods-11-00188-f003:**
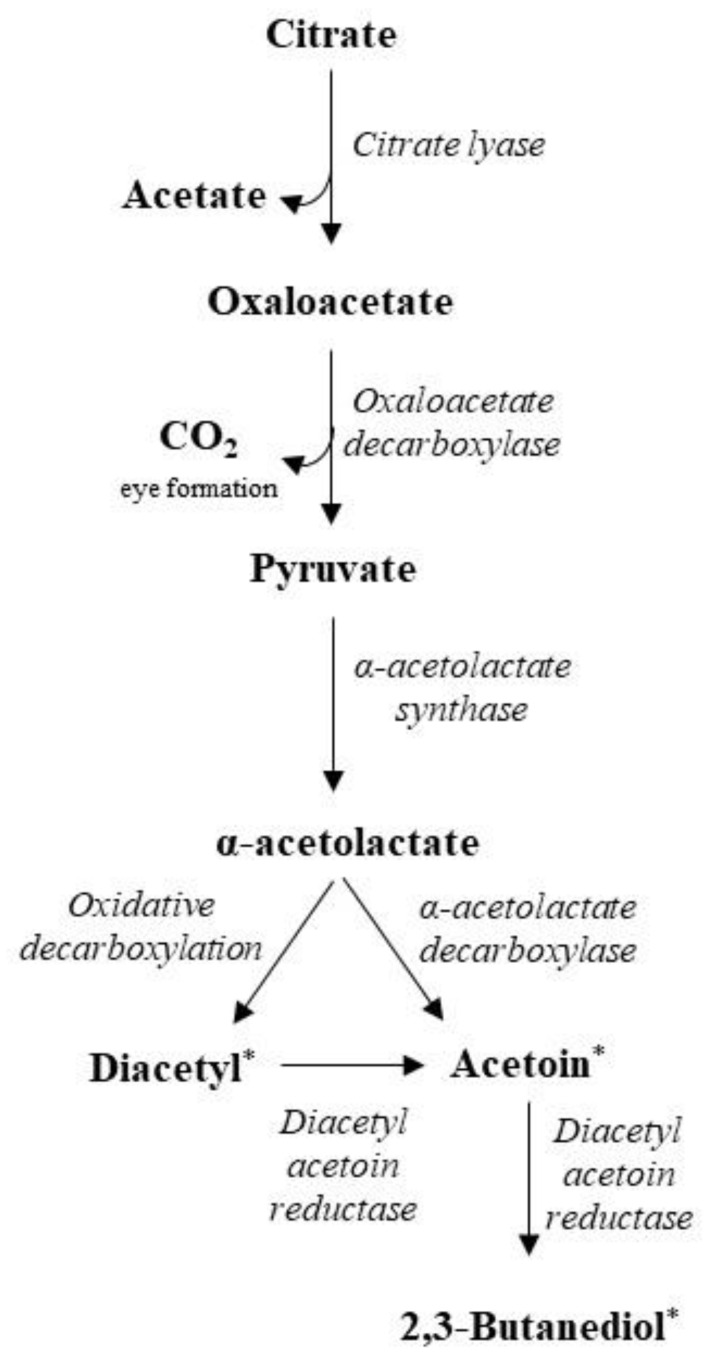
Schematic overview of citrate catabolism by LAB leading to the formation of flavor compounds in cheese. Asterisks denote flavor compounds [6,103,104].

**Figure 4 foods-11-00188-f004:**
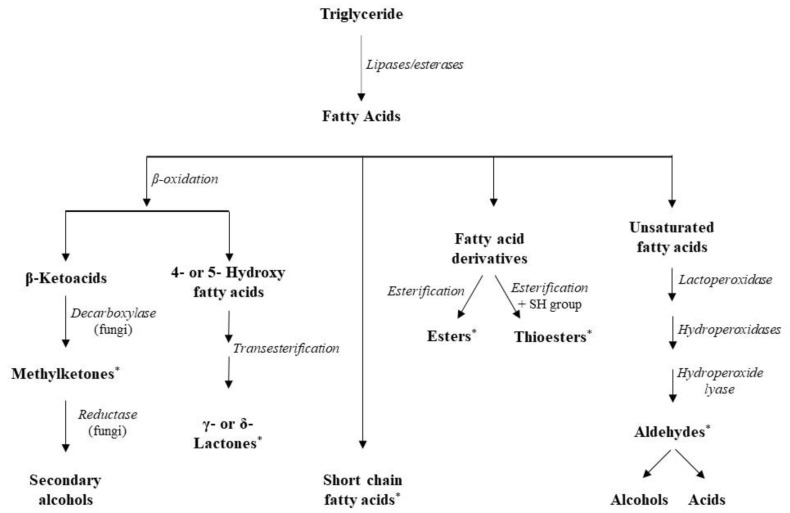
Schematic representation of fatty acids catabolism biochemical pathways leading to flavor compound formation in cheese. Asterisks denote potent flavor compounds [7,95,108,111].

**Figure 5 foods-11-00188-f005:**
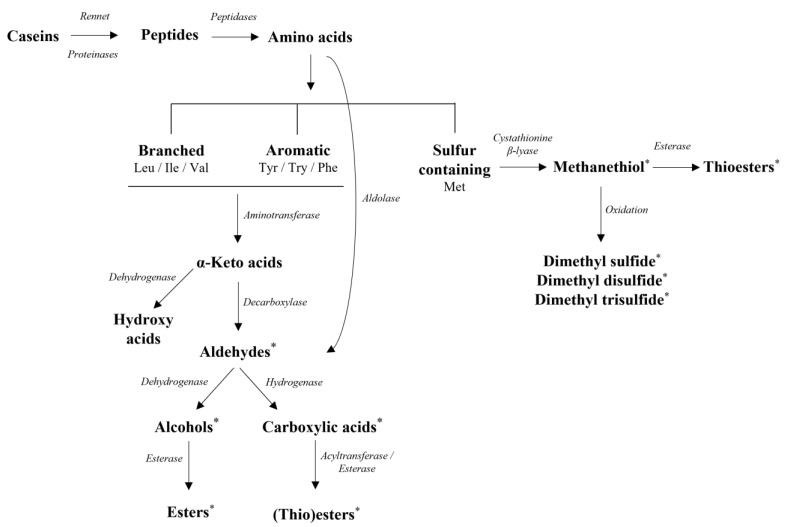
Schematic representation of amino acid catabolism pathways involved in the development of cheese flavor. Asterisks denotes potent flavor compounds [6,103].

## Data Availability

Data is contained within the article or Appendix A.

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
