# Peer review of "Omics Approaches to Assess Flavor Development in Cheese"

_foods, 2022, doi:10.3390/foods11020188_

Round 1
Reviewer 1 Report
Well written review on high-interest topic for microbial community on dairy microbiology. Here follows some suggestions to clarify some points and improve the overall quality of the manuscript.
- line 82: I suggest to put 'cheese production environment' rather than simply 'environment'.
- lines 112-113: I am not convinced that '.. cheese made with pasteurized milk .. without the use of starter cultures ..' exist. I guess about safety features these would be high-risk products. Please verify and give examples.
- line 130: I suggest this change '.. of milk with natural milk or whey cultures consisting of ..'.
- line 137: please add '.. and subsp. lactis' after 'bulgaricus'.
- lines 158-159: definitely, according to authors L delbrueckii and L. helveticus are SLAB or NSLAB (see lines 137-138 and lines 158-159).
- line 180: K. lactis and D. hansenii are yeast not molds of filamentous fungi.
- line 189: spoilage fungi? Not all species of Penicillium genus are spoiling.
- line 275: I suggest this change: '.. nowadays overcome by the application of the so-called -omics approaches'.
- lines 282-283: suggested change 'Their application to cheese, both individually or in combination, will be ..'.
- line 295: add 'contributing to differentiate them' just between cheese and [95].
- lines 312-316: this sentence should be better placed in par. 4.2.
- line 317: put 'becomes the' instead of ' comprises a vital'.
- lines 323-324, suggested change ' .. to assimilate lactose, leading to the accumulation of ethanol and acetic acid [102]'.
- line 442: I suggest to modify the title of this paragraph as follows: 'Omics as a means of deepening the knowledge on flavor formation in cheese'.
- line 706: suggested change '..on the production of volatile compounds during the ..'.
- lines 721-728: too long sentence.
- line 803: what do you mean for 'natural cheese'? It would be better to put 'cheese rinds' rather than simply 'cheese' at the end of line 803.
- line 805: I guess 'to evaluate the microbiota of the cheese samples' is really redundant and could then be erased.
- line 817, change to ' bloomy cheese rinds'.
- lines 847-852: too long sentence.
Author Response
We would like to thank the Reviewer for his/her comments, and we believe that our review article has profited significantly from this revision process.
Please see the attachment with our responses to your comments

Reviewer 2 Report
In this review, the authors introduced the cheese microbiota metabolic pathways during flavor development, and multiple meta-omics approaches assessing flavor development in cheese. The review is generally clear, however, there are some errors throughout the review that detract its overall quality.
L. 24-28. The story is beautiful and attractive. However, the scientific research paper should be based on scientific facts and do not present the content of myth novels, so as to avoid any potential disputes and arguments that may occur in the future. Therefore, I personally suggested to remove these sentences.
L. 76-214. The section of “2. The cheese microbiome” was too large to read, which was difficult to accurately grab useful information in a short time for reader when browsing. It will be much better to add second-level heading.
L. 423, Figure 5. α-keto acids, not a-keto acid.
L. 761. α-tocopherol, not a-tocopherol
L. 431-432. dimethyl sulfide, dimethyl disulfide and dimethyl trisulfide, all three obtained after MTL auto-oxidization. But only “dimethyl disulfide” and “dimethyl trisulfide” were list in the Figure 5. Please add “dimethyl sulfide” in the Figure 5.
L. 443-531. The section of “5.1. (Meta)genomics” contained too much information. It will be much better to be divided into two parts: (1) genetic inheritance: providing genetic information of different strains (bacteria and fungus) during flavor formation by whole genome sequencing, and the distribution of catalytic enzymes in metabolic pathways of flavor formation in different strains (bacteria and fungus), and list a table to make it easier for readers to learn about the metabolic differences of different strains. (2) taxonomical identification: during the flavor formation, the advantages of amplicon sequencing and shotgun metagenome sequencing methods in identifying species (bacteria and fungus), and list examples respectively. And the above two parts add three-level headings respectively.
L. 534. technological changed to genetic.
L. 537. image changed to profile.
L. 533-582. The section of “5.2. (Meta)transcriptomics” contained too much information. It will be much better to be divided into two parts: (1) single microbial transcriptome. (2) comprehensive metatranscriptome.
L. 605-789. The section of “5.4. Metabolomics” contained too much information. It will be much better to be divided into four parts: (1) analytical techniques. (2) cheese-based medium. (3) cheese model system. (4) metabolite profiles.
L. 791-1006. The section of “5.5. Integration” contained too much information. It will be much better to be divided into four parts: (1) amplicon sequencing + metabolomics. (2) shotgun metagenome sequencing + metabolomics. (2) transcriptome + metabolomics. (4) genomic sequencing + transcriptome + metabolomics.
Many cited literatures were older than 5 years.
Author Response

(The authors gave the same response as above.)

Round 2
Reviewer 2 Report
Accept.